# Classification of COVID-19 Patients into Clinically Relevant Subsets by a Novel Machine Learning Pipeline Using Transcriptomic Features

**DOI:** 10.3390/ijms24054905

**Published:** 2023-03-03

**Authors:** Andrea R. Daamen, Prathyusha Bachali, Amrie C. Grammer, Peter E. Lipsky

**Affiliations:** 1AMPEL BioSolutions LLC, Charlottesville, VA 22902, USA; 2RILITE Research Institute, Charlottesville, VA 22902, USA

**Keywords:** COVID-19, severity, classification, transcriptomics, bioinformatics, machine learning

## Abstract

The persistent impact of the COVID-19 pandemic and heterogeneity in disease manifestations point to a need for innovative approaches to identify drivers of immune pathology and predict whether infected patients will present with mild/moderate or severe disease. We have developed a novel iterative machine learning pipeline that utilizes gene enrichment profiles from blood transcriptome data to stratify COVID-19 patients based on disease severity and differentiate severe COVID cases from other patients with acute hypoxic respiratory failure. The pattern of gene module enrichment in COVID-19 patients overall reflected broad cellular expansion and metabolic dysfunction, whereas increased neutrophils, activated B cells, T-cell lymphopenia, and proinflammatory cytokine production were specific to severe COVID patients. Using this pipeline, we also identified small blood gene signatures indicative of COVID-19 diagnosis and severity that could be used as biomarker panels in the clinical setting.

## 1. Introduction

Since its emergence in December 2019, infection by severe acute respiratory syndrome coronavirus 2 (SARS-CoV-2), the causative agent of coronavirus disease 2019 (COVID-19), has led to the deaths of over 6.5 million individuals worldwide [1,2,3]. COVID-19 is characterized by a wide range of disease presentations from mild with flu-like symptoms to severe with acute hypoxic respiratory failure (AHRF) requiring hospitalization and admittance to the intensive care unit (ICU) [4]. In addition, symptoms may persist for months after patients present with acute illness [5]. A number of pre-existing clinical risk factors (age, gender, obesity, respiratory conditions, diabetes, immunodeficiency) [6,7] and post-infection multi-organ disease complications, including cardiac injury, thrombosis, renal disease, and liver injury [8,9,10], have been associated with severe COVID. Despite intense study, however, the major drivers of severe and potentially fatal COVID-19 have not been fully elucidated.

Considerable resources have been applied to study the immune response to SARS-CoV-2 infection as a means to understand COVID-19 pathogenesis in greater detail [11,12]. Among these efforts, our group and others have employed a bioinformatics-based approach using multi-omic, and in particular, transcriptomic data to construct immune profiles of COVID-19 patients at various stages of disease onset and severity [13,14,15,16,17,18,19,20,21,22]. These studies have identified numerous abnormalities in the strength and nature of the adaptive and innate immune responses of severe COVID patients, including neutrophilia, T-cell lymphopenia, plasmablast expansion, antibody or autoantibody production, and IFN levels. With this accumulated knowledge base, the focus has evolved to attempt to apply the information to identify subsets of individuals with COVID-19 with prognostic relevance.

Recent advances in computer science and the use of artificial intelligence (AI) and machine learning (ML) for clinical applications offer a promising approach to identify biomarkers and predict the risk of SARS-CoV-2 infected individuals to develop severe and possibly fatal disease [23,24]. Several recent studies have employed ML or deep learning approaches to patient demographic data [25], lung CT scan images [26,27,28], and transcriptomic data [29,30,31,32,33] to stratify COVID-19 patients with varying degrees of success. However, more work is required to improve model performance and reproducibility before biomarkers of severe COVID-19 identified by ML can be translated into a clinical setting. This is dependent on: (1) developing a standardized approach to dealing with high dimensional data; (2) selecting the best ML algorithm for each application; and (3) selecting the most informative features as input for ML applications. To that end, we have developed an iterative ML pipeline based on curated gene signatures previously used to characterize immune profiles of COVID-19 patients from whole blood gene expression data [13,14]. As a result, we have identified immunological signatures and individual genes with strong predictive capacity for classifying COVID-19 patients most at risk of severe disease that could be employed as a diagnostic and prognostic tool in the clinic.

## 2. Results

### 2.1. A Gene Expression-Based Iterative Machine Learning Approach to Classification of COVID-19 Patients

We employed three publicly available whole blood transcriptome datasets (GSE161731, GSE172114, and PRJNA777938) that had previously been analyzed for immune profiling of COVID-19 patients based on gene signature enrichment by gene set variation analysis (GSVA) [14]. The combined datasets consisted of individuals with COVID-19, individuals with non-COVID-induced AHRF, individuals admitted to the ICU without AHRF, and healthy controls. COVID-19 patients were further categorized as “non-critical” if they exhibited symptoms but were not hospitalized or were admitted to a non-critical care ward and as “critical” if they exhibited more severe symptoms requiring hospitalization and admittance to the ICU. 

To determine the top immune signatures and genes differentiating subsets of COVID-19 patients from healthy individuals and other ICU patients, combined GSVA enrichment scores for 40 curated immune cell and pathway gene signatures [13,14] were used as features to train 9 ML algorithms (Figure 1, Appendix A). Each ML algorithm was used for 4 binary classifications: COVID patients versus healthy individuals, non-critical COVID patients versus healthy individuals, critical versus non-critical COVID patients, and COVID ICU patients versus non-COVID ICU patients. Then, the top 5 performing ML algorithms were employed in an iterative approach to identify the GSVA modules contributing most to each classification (Figure 1). After each iteration, feature importance was calculated for each ML algorithm and the top 50% of features were used for the next round of ML. Then, the log_2_ expression values of genes composing the final 10 gene modules were used as features for ML algorithms to identify the top 20 individual genes that could effectively classify COVID-19 patients in each comparison. Expression values used as input for the final ML iteration were normalized to account for batch effects between separate RNA-seq datasets (Appendix AA). To ensure reproducibility of the results, each ML iteration was repeated 10 times and the most represented modules or gene features were used moving forward. Overall, the ML algorithms successfully performed each binary classification and the top performing algorithms for each comparison achieved accuracies of >0.7 for all iterations of ML. In addition, accuracies improved between the first and last ML iteration as the gene modules were refined and distilled into individual gene features, and final accuracies reached >0.9 for the top 5 algorithms in each classification (Appendix A). Notably, the final gene lists derived from this iterative ML approach reflected distinct immune cell and inflammatory pathways providing the greatest contribution to each binary classification of COVID-19 patients (Appendix A).

### 2.2. Machine Learning Models Reveal Genes Critical for Classification of COVID-19 Patients from Healthy Individuals

The iterative ML pipeline was first applied to determine the top modules and individual genes for classification of COVID-19 patients from healthy individuals. For the initial iteration, support vector machine (SVM) was consistently the top performing algorithm with average areas under the receiver operating characteristic and precision/recall curves (AU-ROC and AU-PR) of 1.0 (Figure 2A) as well as an average overall accuracy of 1.0, indicative of perfect model performance across 10 repetitions (Appendix A). The 10 modules with the highest feature importance were divided between pathways related to cell proliferation/metabolism (Cell Cycle, Pentose Phosphate Signature, Glycolysis, Oxidative Phosphorylation (OxPhos), Fatty Acid Alpha Oxidation (FAAO)) and inflammatory modules (Anti inflammation, Alternative Complement Pathway, Granulocyte, MHC II, Proinflammatory IL-1 Family) (Figure 2B, Appendix A). In the final ML iteration, the perfect classification metrics observed in the first iteration were maintained, but linear regression (LR) and random forest (RF) were the top performing algorithms (Figure 2C, Appendix A). This result highlights the importance of comparing multiple algorithms when inputting different sets of features to the ML pipeline. The final 20 genes with greatest performance in classifying COVID-19 patients from healthy individuals were dominated by genes from modules of the Cell Cycle (CCNB1, CCNB2, CDC20, CEP55, GINS2, MCM10, MKI67, PTTG1) and OxPhos (COX6C, COX7B, COX7C, NDUFAF5, NDUFAF7, NDUFB3, TMEM126B, UQCRB) but also included genes in the Glycolysis (SLC2A3), and Granulocyte (CD177, CLC, OSM) modules (Figure 2D and Appendix A). To determine the exact relationship between expression of these 20 genes and COVID-19 classification, we used SHapley Additive exPlanations (SHAP) values, which provide information on the magnitude and directionality of the contribution of each gene to classification of an individual as a COVID-19 patient or a healthy control. SHAP values are displayed in a summary plot (Figure 2E) in which each dot represents the expression of each gene in an individual patient and higher gene expression values in conjunction with increased SHAP values are indicative of a positive association with COVID-19 classification, and lower gene expression values in conjunction with increased SHAP values indicate a negative correlation with COVID-19 classification. Overall, we found that increased expression of cell cycle and glycolysis genes and decreased expression of OxPhos genes were associated with COVID-19 classification (Figure 2E). Genes in the Granulocyte module were split with increased expression of CD177 and decreased expression of CLC and OSM contributing to COVID classification (Figure 2E).

The COVID-19 patient cohort consisted of individuals with mild, non-critical disease as well as individuals with severe disease requiring ICU admission. Therefore, we sought to determine whether an iterative ML approach could also effectively identify subtle differences between COVID-19 patients with mild disease and healthy individuals (Figure 3). Interestingly, the non-critical COVID vs. healthy classifier performed similarly to classification including COVID-19 patients with severe disease. In the first ML iteration, SVM was the top performing algorithm over 10 repetitions with an average AU-ROC of 0.994, AU-PR of 0.992, and accuracy of 0.976 (Figure 3A, Appendix A). The top 10 gene modules for the non-critical COVID-19 vs. healthy classifier shared common proliferation/metabolism pathways with results when including all COVID-19 patients but, interestingly, had more representation of immune cell gene modules including Plasma Cells (PCs), CD40 Activated B Cells, and Inflammatory Neutrophils (Figure 3B, Appendix A). Algorithm performance improved by the final ML iteration and LR and RF algorithms performed the best with perfect classification metrics in 9 out of 10 repetitions (Figure 3C, Appendix A). The additional immune modules selected as top features identifying non-critical COVID-19 patients from healthy controls were reflected in the final list of genes as in addition to Cell Cycle (CCNB1, CCNB2, CCNE1, CDC20, CEP55, E2F3) and OxPhos (COX7B, COX7C, DNAJC15, NDUFA5, NDUFAF7, TMEM126B, UQCRB), genes in the Inflammatory Neutrophil (CARD16, CFL1, CLEC4D, HBC21), CD40 Activated B Cell (DUSP4), Alternative Complement Pathway (CFD), and Apoptosis (FAS) modules were also included (Figure 3D and Appendix A). SHAP values for the top 20 genes indicated that increased expression of genes in the Cell Cycle, Alternative Complement Pathway, CD40 Activated B Cell, and Inflammatory Neutrophil modules and decreased expression of genes in the OxPhos and apoptosis modules were associated with classification of non-critical COVID-19 patients as compared to healthy individuals (Figure 3E). Thus, an iterative ML approach effectively differentiated COVID-19 patients with varying levels of disease severity from healthy individuals based solely on expression of gene modules and individual genes in each patient.

### 2.3. Iterative Machine Learning Effectively Predicts Disease Severity of COVID-19 Patients Based on Gene Expression

To extend the clinical applications of gene expression-based classification of COVID-19 patients, we employed the iterative ML pipeline to the classification of COVID-19 patients with critical, severe disease requiring hospitalization from patients with a mild, non-critical form of the disease (Figure 4). For the critical vs. non-critical COVID-19 classifier, the gradient boosting (GB) algorithm performed best in the first iteration with an average AUC-ROC and AUC-PR of 0.82 and an average accuracy of 0.83 across 10 repetitions (Figure 4A, Appendix A). The top 10 gene modules distinguishing critical from non-critical COVID-19 largely consisted of inflammatory cell/pathway modules and, in particular, T cells (T Cell, Inflammatory Neutrophil, IFN, Tactivated, Treg, Cytotoxic, Activated T Cell) as well as metabolic pathways (Amino Acid (AA) Metabolism, Pentose Phosphate Signature, FAAO), and the stress response (Unfolded Protein) (Figure 4B, Appendix A). As with the COVID-19 vs. healthy classifier, performance across all algorithms improved by the final iteration and GB remained the top performer with an average AUC-ROC of 0.98, AUC-PR of 0.98, and overall accuracy of 0.93 (Figure 4C, Appendix A). As a result, the majority of the top 20 individual genes identifying critical COVID-19 patients with more severe disease were derived from the Inflammatory Neutrophil module (CARD16, CAST, CD177, FKBP5, H2BC21, HIF1A, MCEMP1, NT5C3A) as well as genes from the IFN (EIF2AK2, SP100), Unfolded Protein (DERL1, ERAP1, SSR1), AA Metabolism (OXCT1), and T-cell-related modules (CCR2, CD28, CREM, EOMES, SGK1) (Figure 4D and Appendix A). SHAP analysis of the directionality of relationships between relative expression of the top 20 genes and classification of critical COVID-19 revealed that, as a whole, critical classification was associated with increased expression of inflammatory neutrophil and IFN genes and decreased expression of T-cell, metabolism, and stress response genes (Figure 4E). These results emphasize the differences in the nature of inflammation present in critical COVID-19 patients as compared to those with mild disease. 

### 2.4. A Gene Expression-Based Machine Learning Approach Identifies Genes Distinguishing COVID-19 ICU Patients from Other Patients Admitted to the ICU

Our previous analysis of patients admitted to the ICU with or without COVID-19-induced AHRF found few differences in gene-expression-based immune profiles, predominantly centered on the nature and magnitude of the PC antibody response [14]. To explore these differences in greater depth, we utilized the iterative ML pipeline for classification of COVID-19 vs. non-COVID-19 ICU patients (Figure 5). The first iteration of this classifier had the weakest initial algorithm performance, and SVM performed best in 5 out of 10 repetitions with an average AUC-ROC of 0.77, AUC-PR of 0.81, and accuracy of 0.76 (Figure 5A, Appendix A). Of the top 10 gene modules for the COVID-19 vs. non-COVID-19 ICU classifier, the PC module had the greatest overall feature importance, as expected from our previous work (Figure 5B, Appendix A). The remaining modules in the top 10 included those related to the inflammatory response (IFN, pDC, CD40 Activated B Cell, LDG, Erythrocytes, TNF) as well as cell proliferation and metabolism (Cell Cycle, TCA Cycle, Pentose Phosphate Signature). Algorithm performance improved dramatically by the final ML iteration in the pipeline, and linear regression (LR) was the best overall algorithm with average AUC-ROC of 0.91, AUC-PR of 0.99, and accuracy of 0.97 (Figure 5C, Appendix A). The top 20 genes predominantly originated from the TNF module (AMPD3, ASAP1, CDKN3, EGR1, FBXL2, FCER2, GMIP, GP1BA) but also included genes from the PC (SDC1), IFN (GBP4, IFITM1, IFITM3), Cell Cycle (MCM10, PTTG1), TCA cycle (CS, SDHC), Pentose Phosphate Signature (G6PD, H6PD), LDG (CAMP), and Erythrocyte (GYPA) modules (Figure 5D and Appendix A). Interestingly, the SHAP summary plot indicated that decreased expression of most of these genes including those from the TNF and metabolism-related modules and increased expression of genes from the PC, IFN, and LDG modules was associated with COVID-19 classification from all patients admitted to the ICU (Figure 5E). Altogether, these results demonstrate that gene expression data can effectively classify subsets of COVID-19 patients and provide insight into inflammatory cells and pathways with the greatest contribution to disease severity.

### 2.5. Validation of the Iterative ML Pipeline in an Independent COVID-19 Patient Dataset

Next, we determined whether the small gene signature output from our iterative ML pipeline could distinguish COVID-19 patients with mild/moderate and severe disease in an independent dataset (GSE157103) [21]. Blood RNA-seq data from 100 hospitalized COVID-19 patients (50 admitted to the ICU and 50 non-ICU) was used to validate the 20 gene signature from the critical vs. non-critical COVID-19 classifier (Figure 4D, Appendix A). Scaled log_2_ expression values for the 20 genes were used as input for 9 ML algorithms and combined ROC and PR curves were generated (Figure 6A). SVM was the top performing algorithm with excellent AUC-ROC and AUC-PRs of 0.97. Classification performance metrics were high for all algorithms with sensitivities ranging from 0.67 to 0.87, specificities from 0.7 to 1, and accuracies from 0.72 to 0.88 (Figure 6B). Therefore, gene signatures derived from the iterative ML pipeline can be successfully applied to classification of COVID-19 severity in an independent patient cohort and achieve a comparable level of model performance.

## 3. Discussion

The heterogeneity of COVID-19 and the wide range of individuals affected by acute and/or persistent disease manifestations necessitates an accurate and reproducible approach to identify drivers of pathology and disease trajectory over time. We have developed a novel, iterative ML approach to identify key molecular signatures stratifying COVID-19 patients based on disease severity and to differentiate them from healthy individuals and those with other infectious or non-infectious causes of AHRF. This information could be leveraged to predict whether an individual that tests positive for COVID-19 will develop mild/moderate disease warranting standard of care treatment or severe disease requiring more careful monitoring and deployment of targeted therapeutics to prevent multi-organ damage and death. Thus, our work is directly applicable to optimizing the deployment of medical resources for COVID-19 patient care. As a result, we identified four separate gene biomarker panels distinguishing COVID-19 patients from healthy individuals, critical from non-critical COVID-19 patients, and critical COVID-19 patients from those admitted to the ICU for non-COVID-19 AHRF. Furthermore, each classifier achieved excellent ML performance metrics across 9 different ML algorithms and over 10 repetitions of each binary classification and effectively translated to an independent validation cohort of COVID-19 patients.

In developing a novel ML pipeline that would yield the most reliable and reproducible results, we considered a number of critical factors including (1) the type of input data to use, (2) the ML algorithm(s) that would be most appropriate, and (3) the method to select the most informative, non-redundant features. First, we used gene expression data as input for the ML pipeline as it produces thousands of data points and, thus, provides a comprehensive overview of COVID-19 disease profiles that is easily obtainable from a single blood draw [34,35,36]. In addition, our previous studies demonstrated the utility of bulk transcriptome data to characterize COVID-19 immunopathology across different tissues and levels of disease activity [13,14], results that have been validated by other multi-omics studies [15,16,17,18,19,20,21,22]. Second, to account for differences and potential biases in ML algorithms used for patient stratification, we utilized 9 different supervised classifiers spanning a wide range of approaches to model construction. These included if/then (DTREE), regression (LR), ensemble (RF, ADB, GBM), Bayesian (NB), dimensionality reduction (LDA), and instance-based algorithms (SVM, KNN) [37,38,39]. In addition, multiple repetitions of each step in the ML pipeline further increased confidence in the reliability and reproducibility of the results. Finally, in working with high-throughput gene expression data, we faced the issue of feature selection and how to reduce transcriptome-wide datasets into a manageable number of gene features that would optimize ML algorithm performance [40,41]. We accounted for this by using curated gene modules representative of immune cells and pathways previously associated with COVID-19 pathogenesis. Then, using an iterative approach in which the most informative features were selected moving forward allowed us to reduce the number of genes used for the final iteration and select the top 20 overall genes able to distinguish COVID-19 patients in each classifier. 

This iterative ML pipeline was used for classification of COVID-19 patients from healthy individuals, critical from non-critical COVID-19 patients, and COVID-19 from non-COVID-19 patients admitted to the ICU with AHRF. Overall, the pathologic cell types and processes represented in the resulting modules and genes reflected previous characterizations of COVID-19 patients, providing validation for these results. Comparing differences in the output of each classifier provided additional insights into the distinguishing features of severe manifestations of COVID-19 as compared to other conditions. Classification of all COVID-19 patients from healthy individuals was dependent on increased expression of genes involved in the cell cycle and decreased expression of genes involved in mitochondrial function through OxPhos. Dysregulated expression of cell cycle genes in COVID-19 patients has been reported in several studies in conjunction with outgrowth of peripheral blood leukocyte populations and thus may serve as a marker of generalized inflammation in infected individuals [42,43,44]. The decreased expression of OxPhos genes is indicative of mitochondrial dysfunction and oxidative stress, both of which are a general feature of the response to viral infection and have also been implicated specifically in COVID-19 patients [45,46,47]. Interestingly, isolating non-critical COVID-19 patients in comparison to healthy individuals yielded a final gene list that, in addition to cell cycle and OxPhos genes, incorporated genes linked with pro-inflammatory cell populations, including inflammatory neutrophils and activated B cells. Our group and others have previously associated neutrophil-driven inflammation and rapid outgrowth of antibody secreting cells (ASCs) with early SARS-CoV-2 infection [14,48], suggesting that these signatures are early signs of mild/moderate disease and that continued expansion of these populations could result in critical illness. 

The final 20 gene list from the critical vs. non-critical COVID-19 classifier indicated that increased expression of inflammatory neutrophil, IFN, and stress response pathway genes in conjunction with decreased T-cell genes were hallmarks of severe COVID-19. Most of the 20 gene signature was composed of inflammatory neutrophil genes, which were originally identified in the blood of severe COVID-19 patients [49,50]. Of these, *CARD16* and *H2BC21* were shared in the signature differentiating non-critical COVID-19 patients from healthy individuals. However, their increased expression in critical patients emphasizes the greater prevalence of pathologic neutrophils in promoting severe disease complications. An elevated IFN response has also been associated with severe COVID-19 in numerous studies in line with a dysregulated systemic inflammatory response and increased expression of stress response genes as tissue damage occurs [16,51,52]. As pro-inflammatory response signatures were increased in severe COVID-19 patients, this was accompanied by a decrease in immunoregulatory population genes and, in particular, T cells indicative of T-cell lymphopenia, which has been frequently described in severe COVID-19 [15,53,54]. Notably, the specific inclusion of cytotoxic-T-cell-specific genes (*EOMES* and *SGK1*) in the 20 gene signature differentiating critical patients further supports a role for inability to clear viral infection in COVID-19 severity.

We also used our iterative ML pipeline to differentiate ICU patients with COVID-19 from those admitted with non-COVID-19 conditions. The most prevalent module for genes resulting from this classifier were indicative of a high TNF response in COVID-19 ICU patients and high expression of TNF family members has been found in damaged tissue of COVID-19 patients indicative of severe disease [55]. This result suggests that while numerous inflammatory cytokines have been implicated in COVID-19 pathogenesis, a TNF response is unique to COVID-19-associated AHRF compared with other conditions requiring admittance to a critical care ward and provides evidence that TNF inhibitors may be a viable option for treatment specifically of COVID-19 [56]. In addition, inclusion of the PC gene *SDC1* in the final gene list distinguishing COVID-19 from non-COVID-19 ICU patients is in agreement with our previous work in which PC enrichment was the major difference in these cohorts [14]. 

Our innovative ML pipeline extends and improves upon previous efforts at ML-based stratification of COVID-19 patients [25,26,27,28,29,30,31,32,33]. A main difference in our approach is the use of whole blood gene expression data, further reduced into groups of gene modules indicative of pathologic cell populations and processes associated with SARS-CoV2 infection and COVID-19 severity. This is in contrast to ML studies utilizing patient demographic/medical record data or lung CT scan images which are limited by the quality and quantity of input data for ML algorithms [25,26,27,28]. Among other ML classification studies using transcriptomic data, our approach is the first to identify the most informative genes from within gene modules previously identified to be abnormally enriched in whole blood of COVID-19 patients. The use of whole blood RNA-seq samples rather than gene expression from sorted single-cell populations [32], peripheral blood mononuclear cells (PBMCs) [32], or throat swabs [31] provided the basis of a clinically useful blood test in which sample handling was minimal. Moreover, this work represents the first study to build transcriptome-based ML models predictive of variations in COVID-19 severity as well as differences from other individuals in the ICU, which were previously assessed separately [29,33].

Overall, our iterative ML pipeline for COVID-19 patient stratification offers a robust method of identifying critical inflammatory processes driving progression to severe disease. In addition, the excellent performance of the resulting algorithms instills confidence that they can be applied to predict disease trajectories of newly diagnosed individuals as demonstrated by the effectiveness in distinguishing severe COVID-19 patients admitted to the ICU from non-ICU COVID-19 patients. We propose that this approach be used for development of rapidly deployable blood-based PCR tests employing a small number of highly informative genes to provide clinical support for COVID-19 patient risk assessment in the clinic.

## 4. Materials and Methods

### 4.1. Study Datasets

Three publicly available whole blood bulk RNA-seq datasets previously analyzed for characterization of COVID-19 patients with varying degrees of severity were used as input for machine learning classification algorithms and an additional dataset was analyzed for validation. Raw SRA files were downloaded from Gene Expression Omnibus (GEO) accessions: GSE161731 [19], GSE172114 [22], PRJNA777938 [14], and GSE157103 [21], converted into FASTQ files, and processed into log_2_ gene expression values as previously described [14].

### 4.2. Gene Set Variation Analysis (GSVA)

The R/Bioconductor package GSVA [57] (v1.36.3) was used as a non-parametric, unsupervised method for determining enrichment of pre-defined gene modules in individual gene expression samples as previously described [14]. GSVA enrichment scores for 40 gene modules (Appendix A) previously used for characterization of immune profiles in blood from COVID-19 patients [13,14] were calculated for each patient and used as input for the iterative machine learning pipeline.

### 4.3. Iterative Machine Learning (ML) Pipeline

The iterative ML pipeline is outlined in Figure 1. Briefly, GSVA enrichment scores for 40 gene modules (Appendix A) from publicly available whole blood bulk RNA-seq datasets were used as input for 9 ML algorithms in each of 4 binary classifications. The pipeline consisted of 3 ML iterations each of which was repeated 10 times. After the first 2 ML iterations, the top 50% of features from the top 5 performing algorithms were used as input for the next iteration. Then, the top 10 gene modules were filtered to remove redundant genes with correlation coefficients >0.8 and log_2_ gene expression values from the remaining genes were normalized across datasets. Normalized z-scores of gene expression values were used as input for the final ML iteration to select the top 20 gene features for each classifier. Detailed methods for each step in the pipeline are described below and available through the following link: https://github.com/adaamen/iterative-ML-COVID-pipeline (accessed on 2 February 2023). 

### 4.4. ML Classification Algorithms

Binary classifications were carried out in Python (v3.8.2) using scikit-learn (v0.24.1) [58]. Nine ML algorithms were implemented to distinguish COVID-19 patients from healthy individuals, critical from non-critical COVID-19 patients, and COVID-19 patients admitted to the ICU from non-COVID ICU patients (logistic regression (LR), random forest (RF), support vector machine (SVM), decision tree (DTREE), AdaBoost (ADB), Gaussian naïve Bayes (NB), linear discriminant analysis (LDA), k-nearest neighbors (KNN), and gradient boosting classifier (GB)). Synthetic minority oversampling technique (SMOTE) [59] was used to account for class imbalances between patient groups used for each classification by expanding the number of samples in the minority class to ensure equal numbers between classes used for ML. For each classification, datasets were split into 70% training and 30% validation and SMOTE was applied to the training data. Classification models for each ML algorithm were then built on the training set using default parameters from the scikit-learn library and evaluated on the validation set. Average algorithm performance was calculated based on the following metrics: sensitivity, specificity, accuracy, and areas under the receiver operating characteristic (ROC) and precision-recall (PR) curves. ROC and PR curves were plotted using the Python library matplotlib (v3.3.4) [60]. Binary classification of ICU and non-ICU COVID patients from an independent dataset were used as validation for results of the iterative ML pipeline. For this classification, SMOTE was not applied. To optimize model performance on the validation dataset, a 75% training, 25% testing split was used and the class_weight = ‘balanced’ parameter was added to the SVM algorithm.

### 4.5. Feature Importance Calculation

Feature importance was calculated for the top 5 performing algorithms in each ML iteration. For each input feature a higher feature importance score indicates how relevant that feature is towards the output of each classifier. The method of calculating feature importance was determined based the appropriate metric for each algorithm and implemented through the scikit-learn Python library. Gini importance, or average decrease in impurity, was used for RF, DTREE, ADB, GB, LR, and LDA, while permutation importance was used for SVM, KNN, and NB.

### 4.6. Gene Feature Correlation Analysis

Feature redundancy was calculated for genes in the top 10 modules of each classifier using Pearson correlations between each feature and every other feature. When two genes shared a correlation coefficient >0.8, the gene with the highest degree of correlation with all other genes in the module was removed. Pearson correlations were computed using the cor() function in R and plotted using the corplot library.

### 4.7. z-Score Gene Expression Normalization

Log_2_ transformed gene expression values were normalized before the final ML iteration to account for batch effects across separate datasets. To do this, z-scores were calculated for each gene across samples in each dataset using the scale() function in R and then normalized datasets were combined into a dataframe used as input for the 9 ML classification algorithms. The pre- and post-normalization expression values for each dataset combination are summarized as boxplots (Appendix A).

### 4.8. SHAP Value Calculation

The precise contribution (magnitude and directionality) of the top 20 gene features output from each classifier was determined using Shapley additive explanations (SHAP) [61]. SHAP values were calculated based on the RF algorithm for the final iteration of each classifier. SHAP summary plots were visualized in Python using the shap library (v0.39.0).

### 4.9. Statistical Analysis

Normalized gene expression values of the top 20 genes were compared between the two classes of each binary classification using multiple unpaired t-tests with Welch’s correction in GraphPad Prism (v9.1.0; www.graphpad.com, (accessed on 20 December 2022)). A two-tailed *p*-value < 0.05 was considered statistically significant.

## Figures and Tables

**Figure 1 ijms-24-04905-f001:**
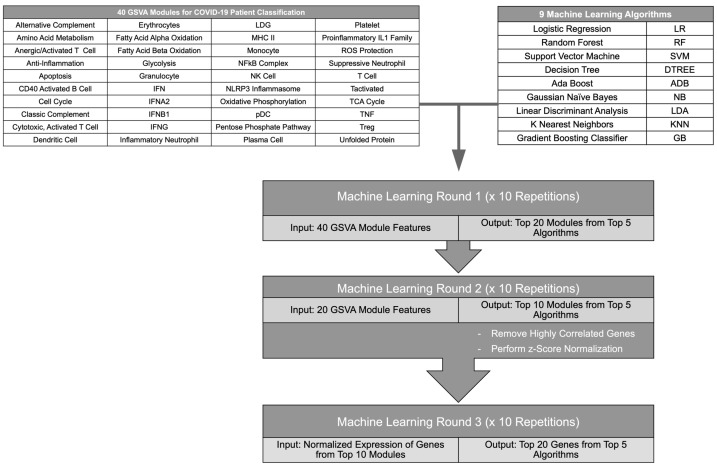
Flow diagram of the iterative ML pipeline. Graphic depicting steps in the iterative ML pipeline used for classification of COVID-19 patients. GSVA scores of gene modules representing immune cells and pathways were used as input for 9 different ML algorithms in 2 iterations after which a final ML iteration used z-score-normalized gene expression values. The flow chart was created in Microsoft Powerpoint.

**Figure 2 ijms-24-04905-f002:**
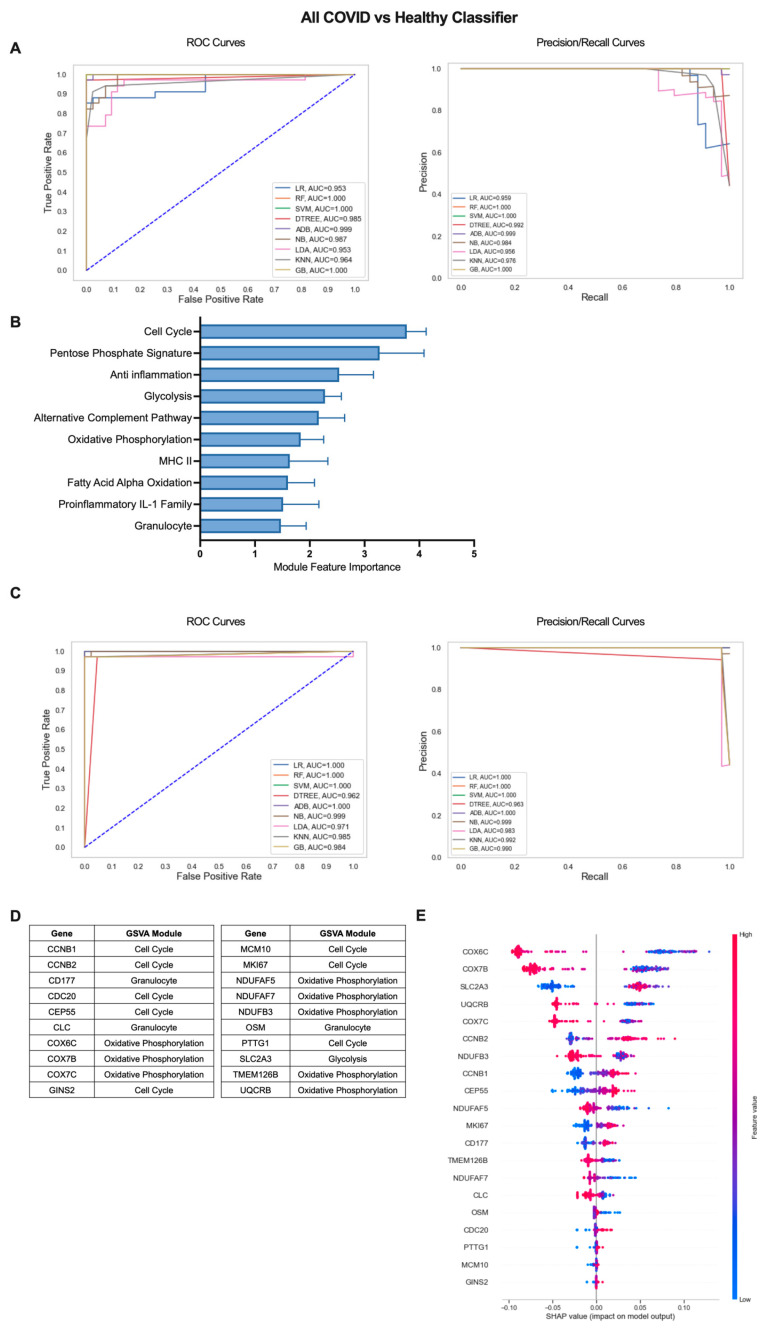
Iterative ML COVID-19 patient vs. healthy individual classifier results. (**A**) Representative ROC and PR curves from the first ML pipeline iteration for the all COVID-19 patients vs. healthy control classifier. (**B**) Bar graph of feature importance values for the top 10 gene modules resulting from the second ML pipeline iteration. (**C**) Representative ROC and PR curves from the third ML pipeline iteration for the all COVID-19 patients vs. healthy control classifier. (**D**) Top 20 gene features resulting from the third ML pipeline iteration. (**E**) SHAP summary plot depicting the magnitude and directionality of relationships between the top 20 gene features and COVID-19 classification.

**Figure 3 ijms-24-04905-f003:**
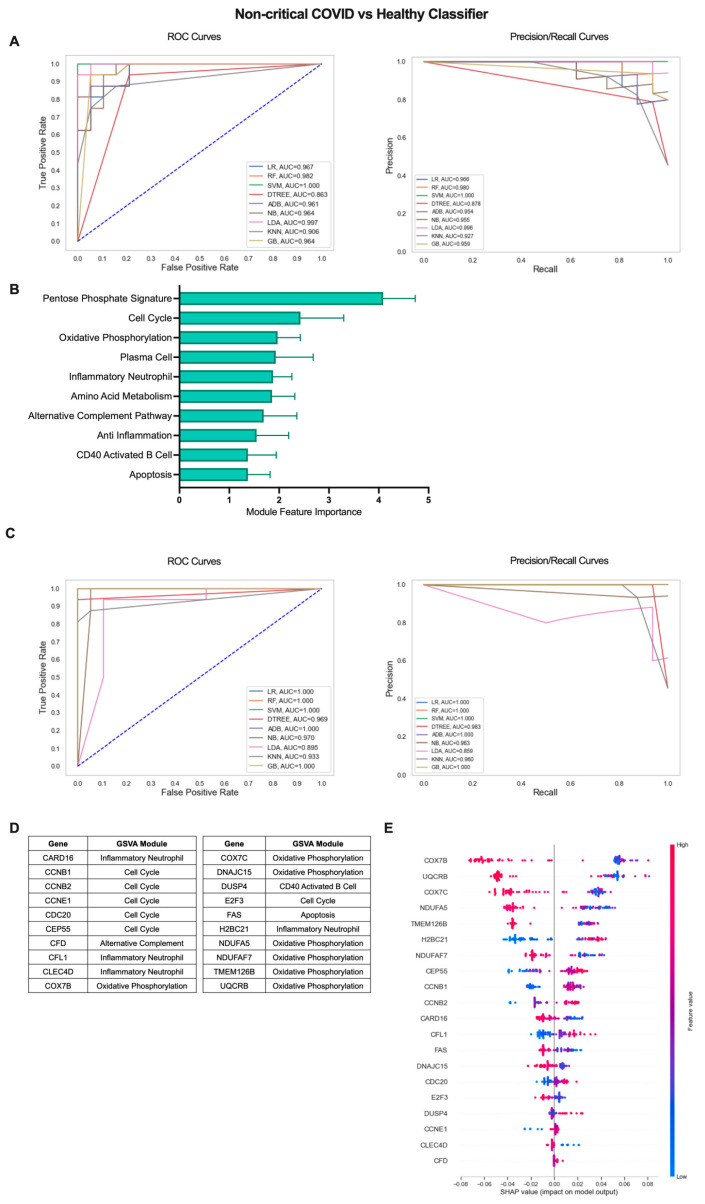
Iterative ML non-critical COVID-19 patient vs. healthy individual classifier results. (**A**) Representative ROC and PR curves from the first ML pipeline iteration for the non-critical COVID-19 patient vs. healthy control classifier. (**B**) Bar graph of feature importance values for the top 10 gene modules resulting from the second ML pipeline iteration. (**C**) Representative ROC and PR curves from the third ML pipeline iteration for the non-critical COVID-19 patient vs. healthy control classifier. (**D**) Top 20 gene features resulting from the third ML pipeline iteration. (**E**) SHAP summary plot depicting the magnitude and directionality of relationships between the top 20 gene features and non-critical COVID-19 classification.

**Figure 4 ijms-24-04905-f004:**
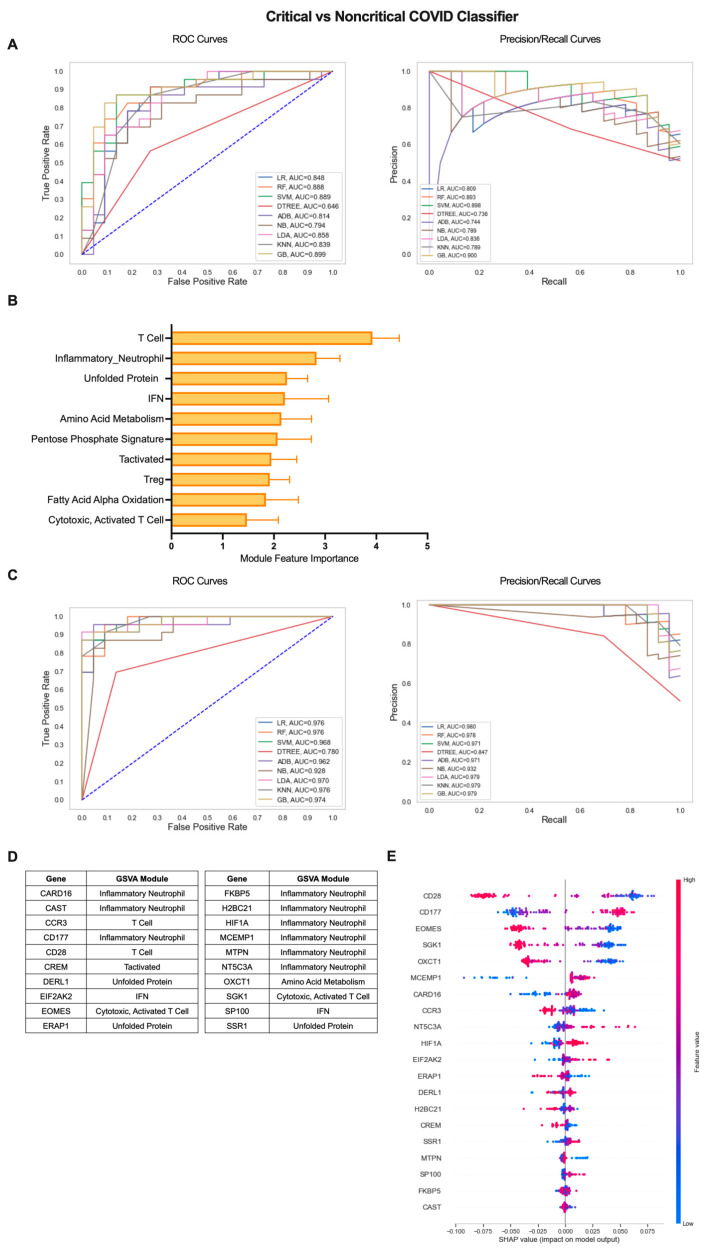
Iterative ML critical vs. non-critical COVID-19 patient classifier results. (**A**) Representative ROC and PR curves from the first ML pipeline iteration for the critical vs. non-critical COVID-19 patient classifier. (**B**) Bar graph of feature importance values for the top 10 gene modules resulting from the second ML pipeline iteration. (**C**) Representative ROC and PR curves from the third ML pipeline iteration for critical vs. non-critical COVID-19 patient classifier. (**D**) Top 20 gene features resulting from the third ML pipeline iteration. (**E**) SHAP summary plot depicting the magnitude and directionality of relationships between the top 20 gene features and critical COVID-19 classification.

**Figure 5 ijms-24-04905-f005:**
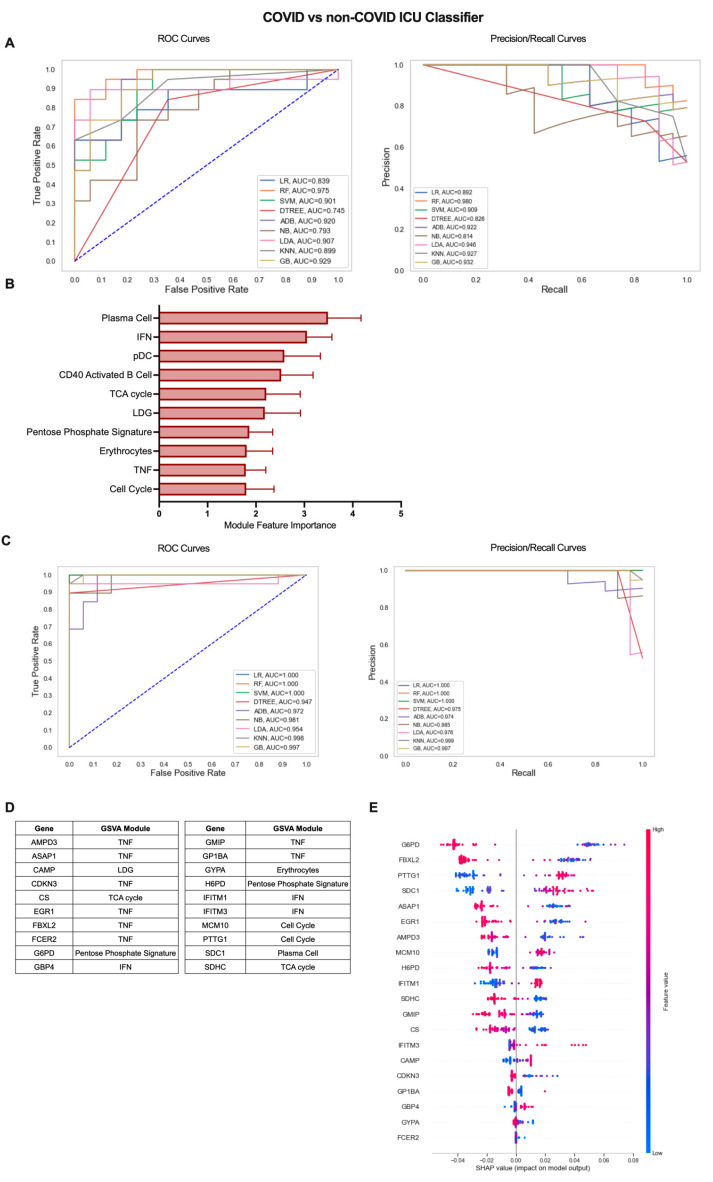
Iterative ML COVID-19 vs. non-COVID-19 ICU patient classifier results. (**A**) Representative ROC and PR curves from the first ML pipeline iteration for the COVID-19 vs. non-COVID-19 ICU patient classifier. (**B**) Bar graph of feature importance values for the top 10 gene modules resulting from the second ML pipeline iteration. (**C**) Representative ROC and PR curves from the third ML pipeline iteration for the COVID-19 vs. non-COVID-19 ICU patient classifier. (**D**) Top 20 gene features resulting from the third ML pipeline iteration. (**E**) SHAP summary plot depicting the magnitude and directionality of relationships between the top 20 gene features and COVID-19 ICU classification.

**Figure 6 ijms-24-04905-f006:**
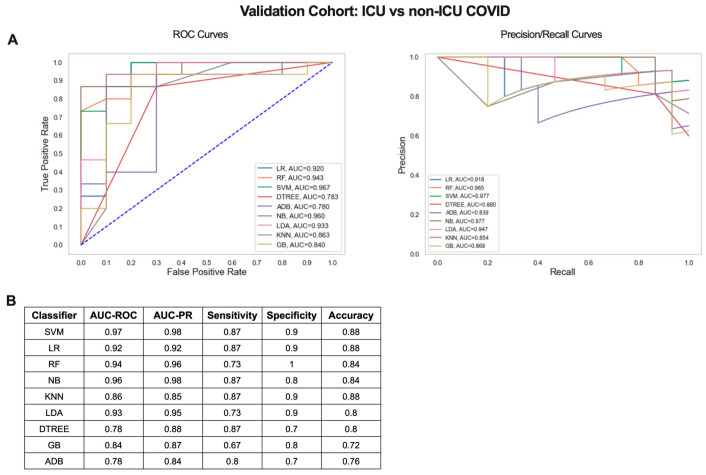
Critical vs. non-critical COVID-19 classifier gene signature validation results. (**A**) ROC and PR curves and (**B**) table of performance metrics for ML results using the 20 gene signature from Figure 4D to compare ICU and non-ICU COVID-19 patients in an independent validation cohort.

## Data Availability

The RNA-seq data analyzed in the current study are publicly available through the National Center for Biotechnology Information (NCBI) under Sequence Read Archive (SRA) project number PRJNA777938 and the Gene Expression Omnibus (GEO) under accessions GSE161731 and GSE172114.

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
