# Peer review of "Classification of COVID-19 Patients into Clinically Relevant Subsets by a Novel Machine Learning Pipeline Using Transcriptomic Features"

_ijms, 2023, doi:10.3390/ijms24054905_

Round 1

Reviewer 1 Report

Daamen et al. used ML approaches to classify COVID-19 patients into clinically relevant subsets by gene expression signature. There are some drawbacks related to study design and reproducibility, which need to be addressed before it considers for publication.

Major comments:

(1) The authors developed a computational pipeline, but scripts and associated data (input data for the scripts) were not provided anywhere. The authors should design a GitHub page/web page where they should upload all the scripts and data that could be used to reproduce the paper results.

(2) The major drawback of this study was that the authors built the model based on log2 gene expression values and did not perform a differential expression analysis. If the identified set of genes is predicted to classify the different clinical signatures of COVID-19 (such as COVID-19 vs healthy and ICU vs non-ICU), in that case, they are expected to be differentially expressed in the corresponding conditions. So, the authors need to perform the DEG analysis of individual datasets and correlate their ML prediction results with the DEGs signature. 

(3) What is the usability and clinical applicability of this pipeline? This should be discussed in more detail. 

Minor comments:

The authors should perform English editing throughout the manuscript. There are several typos and grammatical errors. For example,

In the title, "Relevent" should be "Relevant".

Error in line 400: "COVIVD-19" 

Reviewer 2 Report

The paper introduces an innovative iterative machine learning approach that leverages blood transcriptome data to categorize COVID-19 patients according to the severity of their illness. This pipeline successfully uncovered compact gene signatures from blood samples that can act as biomarkers for COVID-19 diagnosis and severity, which is crucial for accurate prediction of the clinical diagnosis and prognosis of COVID-19 patients. Despite its successes, there are still some open questions that need attention: 

1. The paper title does not highlight the use of iterative machine learning.

2. Figures 2-4 show instances where the AUC value is 1, raising concerns about overfitting.

3. The paper lacks a comparison with the results of other COVID-19 classification studies.

Round 2

Reviewer 1 Report

Although the authors adequately addressed all of my comments, I still have a concern regarding the applicability of their method; because they did not provide their scripts and proper guidance for using their pipeline. A proper GitHub page with all the important scripts and detailed descriptions could greatly enhance the applicability of their method.

Author Response

In response to the reviewer's request, we have created a Github page containing the methods and scripts for the iterative ML pipeline described and implemented in this work. We have also added a link to the page in the Methods section of the manuscript text.